# On the Adsorption of Cerium(III) Using Multiwalled Carbon Nanotubes

**Francisco José Alguacil** [ID]**, Irene García-Díaz, Esther Escudero Baquero, Olga Rodríguez Largo and Félix Antonio López \*** [ID]

TecnoEco Group, Department of Primary Metallurgy and Materials Recycling,
Centro Nacional de Investigaciones Metalurgicas (CSIC), Avda. Gregorio del Amo 8. 28040 Madrid, Spain;
fjalgua@cenim.csic.es (F.J.A.); irenegd@cenim.csic.es (I.G.-D.); mebaquero@cenim.csic.es (E.E.B.);
olga.rodriguez@csic.es (O.R.L.)
\* Correspondence: f.lopez@csic.es; Tel.: +34-915538900

**Abstract:** Commercially available oxidized (carboxylic groups) and nonoxidized multiwalled carbon nanotubes were studied as adsorbents of cerium(III) in batch operation mode. Several variables affecting the rare earth adsorption were investigated, including: the stirring speed applied to the system, the pH of the solution, and the metal concentration and carbon dosages. Although the removal of cerium from the solution is different and dependent upon the adsorbent type—(i) adsorption in nonoxidized multiwalled carbon nanotubes, (ii) cation exchange in the case of using oxidized multiwalled carbon nanotubes—the adsorption kinetics, the rate law and the isotherm models are the same for both adsorbents: pseudo-second order, film diffusion, and Langmuir Type-1, respectively. Cerium is desorbed from loaded adsorbents using acidic solutions.

**Keywords:** adsorption; carbon nanotubes; rare earth

## 1. Introduction

In comparison with other separation technologies, adsorption is one of the most used due to its operational characteristics and the possibility of its uses in dilute solutions and even nonclarified ones [1]. Thus, from a long time ago, adsorbents of various natures are used as a medium to remove both valuable and nonvaluable, including hazardous, metals or solutes from aqueous solutions [2,3]. Including these adsorbents, carbon nanotubes in various configurations are considered of a particular interest in this task [4–7].

Among these applications is the case of the recovery of rare earths from various aqueous solutions [8,9]. Nowadays, with a global mine production of $2.1 \times 10^5$ ton in 2019 [10], the recovery of these rare earths is of particular interest due to their use in smart technologies and scarce availability of their resources (i.e., here in the European Union); thus, the recycling of these metals became of interest. However, in the European Union, the rate of their recyclability is around 6% and 7% for the heavy and light rare earths, respectively [11,12].

Cerium and cerium compounds are of particular importance due to their application in strategical sectors, including metal alloys, glass, adsorbent, catalysts, biomedical applications [13–20], and also in a growing sector such as rechargeable batteries [21–23]; thus, cerium is considered a critical raw material by the EU [24].

Recent investigations on the adsorption of cerium(III) showed that a variety of adsorbents is useful for this task, i.e., K10-montmorillonite removes Ce(III) from solutions via a cation exchange mechanism [25], whereas simulated radioactive Ce(III) can be removed from solutions using magnetic trioctylamine–polystyrene composite microspheres [26]. Nitrolite is used as an adsorbent for a series

of light rare earths and Cr(III) [27]; in all the cases, the adsorption increases with the increase in the pH (up to pH 9); however, metal precipitation phenomena occur together with a true adsorption process. Other adsorbents recently used for the removal of cerium(III) from aqueous solutions are various soils [28], montmorillonite nanoclay [29], HKUST-1 framework [30], and chert rocks [31] in this work, and being one of the rare exception in adsorption experiments, the stirring speed applied to the system is considered. Relying on not a true adsorption process but on an ion exchange process, several ion exchangers resins (Amberlite 200C Na, Amberlite 200C, Dowex M4195 and Diphonix) are used to investigate the recovery of Ce(III) (among other metals) from acidic solutions [32,33].

From several years ago, we have used the two particular adsorbents mentioned below in the removal of several metals from aqueous solutions [4,34–38]. The reason is that both materials are easily accessible in the market and, based in our experience, represented good alternatives with respect to other adsorbents, at least in terms of availability and price. Continuing in our research efforts, the present investigation shows an experimental work on the adsorption of cerium(III) from aqueous solutions using two carbon nanomaterials: nonoxidized multiwalled carbon nanotubes (MWCNTs) and oxidized multiwalled carbon nanotubes (ox-MWCNTs); in this case, the material has been functionalized by carboxylic acid groups. Several experimental variables affecting cerium(III) adsorption were investigated, and the data were fitted to several models to explain the kinetics, rate law, and adsorption isotherms associated with this adsorption. Metal loaded onto both nanomaterials can be desorbed by the use of acidic solutions.

## 2. Materials and Methods

The adsorbents were used as supplied by the vendor (Merk KGaA, Damstadt, Germany); in the case of the oxidized material, carboxylic acid was the active group. Main characteristics of both adsorbents were given elsewhere [4,34]. On both carbonaceous nanomaterials, Z potential was measured using a Zetasizer Malvern Nano ZS (Malvern Panalytical, Malvern, UK) at 25 °C. Aqueous suspensions were prepared in the 1–13 pH range, using HCl and NaOH solutions to control the pH value. The concentration of the carbon material was adjusted to a value of 0.1 g/L. The suspensions were dispersed with a Bandelin Electronic Sonopuls HD 3100 sonicator (Bandelin, Berlín, Germany), using an amplitude of 60% for 150 s.

Other chemicals used were of AR grade. Cerium(III) solutions were prepared by dissolving $Ce(NO_3)_3 \cdot 6H_2O$ (Merk KGaA, Damstadt, Germany) in distilled water, and the different concentrations used in the experiments were prepared from this stock solution. The pH values of the aqueous solutions were adjusted by the addition of nitric acid. During the experiments, the pH was continuously monitorized using a 605 pH-meter (Hach Loveland, Colorado, CO, USA).

The adsorption experiments were carried out in a 250 mL glass vessel, fitted with mechanical stirring (Heidolph Instrument GmbH & Co. KG, Schawabach, Franconia, Germany) via a four blade (2.6 cm diameter) glass impeller. Cerium concentrations in the aqueous solution were analyzed by inductively coupled plasma atomic emission spectroscopy (ICP-OES) using Agilent 5100 (Agilent Technologies, Santa Clara, CA, USA), with an associated analytical error of ±3%. The percentage of cerium remaining in the aqueous solution was determined by the following relationship:

$$\% \ = \ \frac{[Ce]_{aq,t}}{[Ce]_{aq,0}} \times 100 \tag{1}$$

where $[Ce]_{aq,0}$ and $[Ce]_{aq,t}$ were the initial and at an elapsed time cerium concentrations in the solution, respectively. Cerium concentrations in the adsorbents were calculated by the mass balance. Desorption experiments were performed in the same basis as above.

## 3. Results

### 3.1. Estimation of the Isoelectronic Points

Measurement of $Z$ potential values on MWCNTs and ox-MWCNTs, at different pH values, showed that the isoelectronic points (IEP) were 1.22 and 0.26 for both nanomaterials, respectively. At pH values lower than the pH(IEP), the charge of the adsorbent surface was positive; however, when the pH increased, from the respective isolecetronic point value, the surface had a negative charge. This change in the surface charge may have, as it is demonstrated in the latter, a great influence on the adsorption results.

### 3.2. Adsorption Using Nonoxidized Multiwalled Carbon Nanotubes (MWCNTs)

The adsorption of cerium(III) by MWCNTs was represented by the general equilibrium:

$$Ce_{aq}^{3+} \leftrightarrow Ce_c^{3+} \tag{2}$$

where aq and c denoted the species in the aqueous and carbon phases, respectively. Cerium(III) loaded onto the carbon due to the attraction between the negatively charged carbon surface and the positively charged metal ion.

#### 3.2.1. Influence of the Stirring Speed

Experiments to investigate the influence of the stirring speed applied on the system were first investigated by the use of aqueous solutions of 0.01 g/L Ce(III) at pH 6 and a carbon dosage of 4 g/L. In these series of experiments, the stirring speed was varied from 500 to 1000 min$^{-1}$, whereas the temperature was maintained at 25 °C. The results were summarized in Table 1.

**Table 1.** Influence of stirring speed on cerium adsorption.

| Stirring Speed, min$^{-1}$ | Ce(III) Adsorption, % |
|---|---|
| 250 | 90 |
| 500 | 99 |
| 750 | 98 |
| 1000 | 96 |

Time: 5 h.

It can be seen that cerium uptake, except at 250 min$^{-1}$, became practically independent of the stirring speed, though a slight decrease in metal uptake was observed from 500 to 1000 min$^{-1}$, probably due to the presence of local equilibria produced by the increase in the centrifugal force as the stirring speed increased. These results demonstrated that around 500 min$^{-1}$, a minimum value of the thickness of the aqueous phase boundary layer was reached.

The experimental results yielded at this stirring speed were used to estimate the kinetic order associated with this system, and the results from this fit resulted in the cerium adsorption by these MWCNTs that was best represented by the pseudo-second order kinetic model ($r^2 = 0.9982$) [34]:

$$\frac{t}{[Ce]_{c,t}} = \frac{1}{k[Ce]_{c,e}^2} + \frac{1}{[Ce]_{c,e}}t \tag{3}$$

In the above Equation, $[Ce]_{c,t}$ and $[Ce]_{c,e}$ were the cerium concentrations in the carbon at an elapsed time and at equilibrium, respectively, and k was the rate constant associated with the model. The fit indicated that k was estimated as 0.38 g/mg min, and $[Ce]_{c,e}$ was 2.6 mg/g, which fit very well with the value of 2.5 mg/g experimentally yielded. Since these experimental data fitted with the pseudo-second order model, a chemical activation between MWCNTs and the cerium ions might be possible [39].

### 3.2.2. Influence of Aqueous pH

Table 2 shows the results obtained from the adsorption of cerium at different aqueous pH values. The carbon dosage was of 3 g/L. The aqueous solution contained 0.01 g/L Ce(III) at pH values ranging from 1 to 6.

**Table 2.** Influence of the aqueous pH.

| pH | Ce(III) Adsorption, % |
|----|------------------------|
| 1  | 1   |
| 2  | 10  |
| 3  | 29  |
| 4  | 78  |
| 5  | 85  |
| 6  | 99  |

Temperature: 25 °C. Time: 5 h. Stirring speed: 500 min$^{-1}$.

The results shown in Table 2 demonstrate that the increase in the aqueous pH increased the percentage of metal uptake onto the adsorbent. Therefore, these results agreed with the isoelectronic point of this adsorbent (1.22), and since the pH increased from this point, the carbon surface became negatively charged, thus promoting cation uptake. It should be noted here, that at pH values greater than 8, cerium(III), as most Rear Earth Elements (REEs) do, precipitates.

The results at pH 6 were used to estimate the rate law explaining cerium adsorption onto these nanomaterials. The results from this fit demonstrated that aqueous diffusion controlled such uptake ($r^2 = 0.9815$) [34]:

$$ln(1 - F) \ = \ -kt \tag{4}$$

where *F* represented the factorial approach to the equilibrium, defined as:

$$F \ = \ \frac{[Ce]_{c,t}}{[Ce]_{c,e}} \tag{5}$$

Accordingly, with Equation (4), the rate constant k for the model was estimated as 0.02 min$^{-1}$. It is worth to mention here, that in a series of adsorption systems the pseudo-second order kinetic model was associated with an aqueous diffusion model [40].

### 3.2.3. Effect of the Initial Metal Concentration on Cerium Uptake

The initial metal concentration in the solution is considered as a key driving force to overcome the metal mass transfer resistance between the adsorptive material and aqueous phases.

Table 3 shows the variation in the percentage of cerium adsorption against the concentration of cerium ranging from 0.01 to 0.08 g/L in the aqueous solution. These results show that Ce$^{3+}$ adsorption is dependent on the initial metal concentration, since it supplies the major driving force for overcoming limitations of mass transfer between the multiwalled carbon nanotubes and the aqueous phases.

It can be observed that with the present experimental conditions, the percentage of metal adsorption onto the multiwalled carbon nanotubes tended to decrease because the quantity of the accessible sites of adsorption on the surface of the nanotubes decreased and, therefore, their saturation took place easily. As a result, by occupying valence forces via the attraction of the opposite charges of the carbon nanotubes and the metal ion, more mass transfer from the aqueous solution occurred, which was estimated as a slow process [41]. The data derived from Table 3 were used to estimate the

isotherm model which best fitted to the adsorption results, the results from this fit indicated that that Langmuir Type-1 isotherm presented the best fit ($r^2$ = 0.9987) to these experimental data:

$$\frac{[Ce]_{aq,e}}{[Ce]_{c,t}} = \frac{1}{k[Ce]_{c,m}} + \frac{1}{[Ce]_{c,m}}[Ce]_{aq,e} \tag{6}$$

where $[Ce]_{aq,e}$ represented the metal concentration in the aqueous phase at the equilibrium, and $[Ce]_{c,m}$ of the maximum cerium concentration adsorbed into the nanotubes. From the fit, it was derived that $[Ce]_{c,m}$ was 10 mg/g, a value which coincided with the experimental one, whereas k was of 0.15 g/L. From the next expression:

$$R_L = \frac{1}{1 + k[Ce]_{aq,0}} \tag{7}$$

it was estimated that $R_L < 1$, indicating that metal uptake was a favorable process for all the cerium concentrations used in this work [42].

**Table 3.** Variation in the metal adsorption with the initial cerium concentration.

| $[Ce]_{aq,0}$, g/L | Ce(III) Adsorption, % |
|---|---|
| 0.01 | 99 |
| 0.02 | 78 |
| 0.03 | 77 |
| 0.04 | 76 |
| 0.06 | 52 |
| 0.08 | 38 |

Aqueous phase: Ce(III) at pH 6. Carbon dosage: 3 g/L. Temperature: 25 °C. Time: 5 h.

### 3.2.4. Effect of the Dosage Absorbent on Cerium Adsorption

In Table 4, the obtained percentage of cerium adsorption at various MWCNTs dosages were given.

**Table 4.** Cerium adsorption vs. multiwalled carbon nanotube (MWCNT) dosage.

| Nanotubes Dosage, g/L | Ce(III) Adsorption, % |
|---|---|
| 1 | 0.5 |
| 2 | 28 |
| 3 | 99 |
| 4 | 99 |

Aqueous solution: 0.01 g/L Ce(III) at pH 6. Temperature: 25 °C. Time: 5 h. Stirring speed: 500 min$^{-1}$.

Results obtained revealed that there was an increase in the percentage of cerium adsorption with the increase in the adsorbent dosage up to 3 g/L, and then no significant change in this percentage at higher adsorbent dosage. The reason was that the cerium ions competed for confined adsorption sites at a lower adsorbent dosage. However, by increasing the adsorbent dosage, a greater surface area and more empty sites were available; therefore, adsorption efficiency increased until an excess in the adsorbent dosage occurred, i.e., at 4 g/L in the present case, and the efficiency remained constant [43].

### 3.3. Adsorption Using Oxidized Multiwalled Carbon Nanotubes (ox-MWCNTs)

Having this adsorbent carboxylic acid as functional groups, it was logical to attribute metal uptake by the adsorbent to the following equilibrium:

$$3(R-COOH)_c + Ce^{3+}_{aq} \leftrightarrow (R-COO^-)_3Ce^{3+}_c + 3H^+_{aq} \tag{8}$$

In the above Equation, c and aq represented the carbon and aqueous phases, respectively, and R represented the nanotubes matrix. Accordingly, cerium uptake by these adsorbents was attributed to a cation exchange process, in which the metal uptake onto the nanotubes was accompanied by the release of protons to the aqueous solution.

### 3.3.1. Influence of the Stirring Speed

Experiments were carried out to establish the adequate hydrodynamic conditions in the system. Results obtained are shown in Figure 1. Near constant adsorption for stirring speeds in the 500–1000 min$^{-1}$ was obtained; thus, the thickness of the aqueous diffusion layer and the aqueous resistance to mass transfer were minimized.

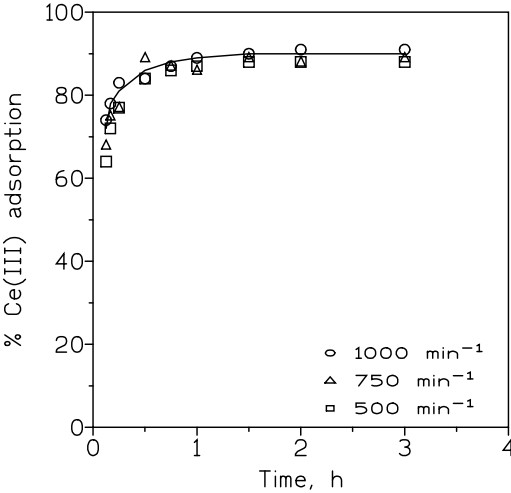

**Figure 1.** Influence of the stirring speed on the adsorption of cerium(III). Aqueous phase: 0.01 g/L Ce(III) at pH 6. ox-MWCNTs dosage: 1 g/L. Temperature: 25 °C.

Experimental results were used to estimate the kinetics model which fitted to these values. These calculations showed that the pseudo-second order kinetic model (Equation (3)) best fitted ($r^2 = 1.0$) to the experimental results; from this fit, the equilibrium cerium concentration in the carbon material was of 9 mg/g (a value which fitted perfectly with the experimental one), whereas the rate constant for the model was of 100 g/mg min.

### 3.3.2. Influence of the Initial Cerium Concentration

A series of tests was performed using aqueous solutions with different contents of cerium at pH 6. The adsorbent dosage was of 0.5 g/L.

Table 5 shows the variation of cerium adsorption and metal uptake onto the adsorbent for the different metal concentrations. This shows that the increase in the initial cerium concentration tended to decrease the percentage of metal adsorption.

**Table 5.** Influence of initial cerium concentration in metal adsorption.

| $[Ce]_{aq,0}$, g/L | Ce(III) Adsorption, % |
|---|---|
| 0.01 | 92 |
| 0.02 | 92 |
| 0.03 | 77 |
| 0.04 | 61 |
| 0.06 | 39 |
| 0.08 | 30 |

Temperature: 25 °C. Time: 5 h. Stirring speed: 500 min$^{-1}$.

Different isotherm models were used to estimate their fit to the experimental values derived from the data showed above. The results from this fit indicated that the cerium uptake was best represented ($r^2 = 0.9997$) by the Langmuir Type-1 isotherm (Equation (6)), resulting in a k value of 3 L/mg and $[Ce]_{c,m}$ of 48 mg/g, a result which coincides with the experimentally derived value. Using Equation (7), it was found that $R_L < 1$ was indicative of a favorable exchange process.

### 3.3.3. Influence of the pH

The variation in the adsorption, as a function of the aqueous pH value, at 0.5 g/L carbon nanotubes dosage, was investigated. The experimental conditions were: aqueous phases containing 0.01 g/L Ce(III) at various pH values; temperature—25 °C, time—5 h; and stirring speed of 500 min$^{-1}$.

Results are shown in Figure 2, in which a decrease in cerium adsorption occurred as the pH of the solution decreased. The results can be explained according to what is shown in Equation (8); that is, a shift of the equilibrium to the left as the proton concentration in the solution increased.

From this Equation (8):

$$K = \frac{[Ce]_c[H^+]_{aq}^3}{[ox-MWCNTs]_c[Ce^{3+}]_{aq}} \tag{9}$$

and:

$$\log D = \log C + 3\,pH \tag{10}$$

considering:

$$D = \frac{[Ce]_c}{[Ce^{3+}]_{aq}} \tag{11}$$

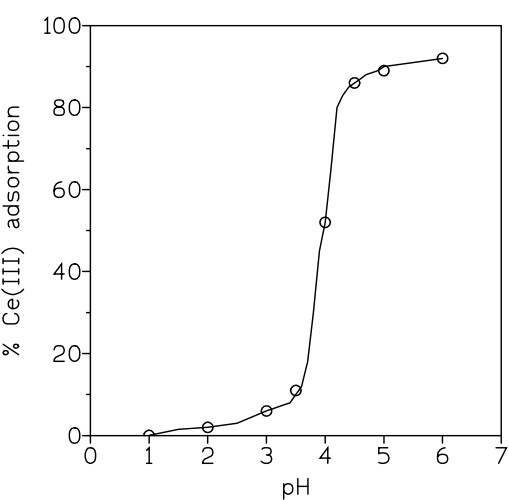

**Figure 2.** Plot of Ce(III) adsorption (%) vs. pH.

In the above Equation (9), the subscripts c and aq denoted the carbon and aqueous phases, and the term $\log C$ represented the other terms as in Equation (8). Thus, from Equation (10), a plot of $\log D$ vs. pH might result in a line with slope 3; from results presented in Figure 2, such plot effectively had a slope near 3.

The results at pH 6 were used to estimate the rate law governing the adsorption of cerium onto the nanotubes. Experimental data best fitted ($r^2 = 0.9348$) to the intraparticle diffusion model:

$$\ln\left(1 - F^2\right) = -kt \tag{12}$$

where $F$ was defined as in Equation (5). The value of the rate constant $k$ was estimated as 0.098 min$^{-1}$.

### 3.3.4. Influence of the of the Adsorbent Dosage on the Adsorption of Cerium

The adsorption of Ce(III) solutions (0.01 g/L at pH 6) using different adsorbent dosages (0.13–1 g/L) was investigated. Experimental results revealed no change in the percentage of cerium adsorption (92%) at higher adsorbent dosages (0.5–1 g/L), whereas these yields were of 31% and 60% using dosages of 0.13 and 0.25 g/L, respectively. In these series of experiments, other experimental variables were fixed as in Table 5.

### 3.4. Desorption

As shown in previous subsections, the pH of the aqueous solution has a great influence on cerium(III) adsorption; thus, the desorption step was investigated by the use of acidic solutions as desorbents on both nanomaterials. The experiments were carried out at 25 °C using 0.1 M $H_2SO_4$ solutions and a Ce-loaded carbon nanomaterial in a 0.4 carbon/solution relationship. From these experiments, it was shown that the percentage of cerium desorption was 75% and 84% in the case of the nonoxidized and the oxidized nanomaterials, respectively. Together with the recovery of the metal, the carbon nanotubes were recovered for further use. In the case of the nonoxidized MWCNTs, the use of the nonconcentrated acidic solution activated the adsorption sites present in the nanomaterial matrix without destroying it. However, and as shown in the literature [44], different behavior between the initial and eluted carbon material might be observed under continuous use, i.e., columns. In the case of the oxidized nanotubes, this regeneration was caused by shifting to the left of the equilibrium, showed in Equation (8), as a consequence of the desorption reaction.

With respect to the above results, the use of HCl or $HNO_3$ solutions, as desorbents, does not improve the respective percentages of cerium desorption.

## 4. Conclusions

Experimental results showed that it is possible to use both commercial available nanomaterials to adsorb Ce(III) from aqueous solutions.

For both nanomaterials, the adsorption of cerium, is influenced by a number of variables, including, metal concentration, adsorbent dosage, and the pH of the aqueous solution. However, the stirring speed applied on the system has near a negligible influence on cerium uptake.

Using both adsorbents, the metal uptake responded to the pseudo-second order kinetic model and the Langmuir Type-1 isotherm model; however, they differed in the rate law, as the film diffusion model fitted the uptake when the nonoxidized material was used, against the fitting of the particle diffusion model to the results yielded when the oxidized carbon nanotubes were used to remove cerium from the solution. The mechanism from which cerium is removed from the solution, using both nanomaterials, also appeared to be different: an adsorption process in the case of the nonoxidized nanomaterial against a cation exchange process in the case of the oxidized material. This is somewhat reflected in the fact that to obtain the same degree of cerium removal from the solution (more than 90%), the material dosage using the oxidized material is three times lower than that of the nonoxidized carbon nanotubes. Cerium loaded onto both nanomaterials can be desorbed by the use of sulphuric acid solutions.

**Author Contributions:** Conceptualization, F.J.A. and F.A.L.; methodology, F.J.A. and F.A.L.; formal analysis, I.G.-D. and F.J.A.; investigation, F.J.A., F.A.L., I.G.-D., E.E.B. and O.R.L.; resources, F.A.L.; writing—original draft preparation, F.J.A.; writing—review and editing, F.J.A., F.A.L., I.G.-D., E.E.B. and O.R.L. All authors have read and agreed to the published version of the manuscript.

**Funding:** This research has received funding from the European Union's Horizon 2020 research and innovation program under grant agreement No 776851 (CarEService).

**Conflicts of Interest:** The authors declare no conflict of interest.

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
