# Peer review of "On the Adsorption of Cerium(III) Using Multiwalled Carbon Nanotubes"

_metals, doi:10.3390/met10081057_

Round 1

Reviewer 1 Report

The paper is well written and is interesting, however the results lack some discussion.

More specific comments were made in the pdf file attached.

Reviewer 2 Report

Dear authors!
Your work on the adsorption of cerium(III) may be of interest to the scientific community. However, I have some comments and questions to consider.

Introduction:
1. In this section you should write about the adsorbents that you use. What metals have they been used before to remove? Are they effective? What are their features? Why don't you consider adsorption as a separation method and evaluate the effects of other metals? Or can you offer a real solution that contains only cerium(III)?

2. You need to explain to the reader your decision to use these adsorbents. Why are they better than others? You need to make the reader more clear about the relevance of the work.
3. line 50 - Do you need the word "and"?
4. line 56 - the two dots at the end of the sentence.

Materials and methods:
1. To describe the reagents and equipment, add the following: "Company name, city, state, country."
2. What kind of stirring equipment did you use? Give the relevant information.

Results:
1. Lines 79-83 - The line spacing is different from the text.
2. In section 3.1.1. what is the time of the experiment? Why is such a small range taken? It is necessary to increase, starting from the minimum.
3. Make sure that all the formulas are in the same format.
4. Section 3.2.2. repeat itself twice. Be more careful!
5. Lines 193-195 - correct indent.
6. Point 243 - "literature38." - 38 is that a reference? If so, it's not on the References!

Reviewer 3 Report

attachment
